ROM-Pose: restoring occluded mask image for 2D human pose estimation

Lee Yunju 1
http://orcid.org/0000-0003-2358-4021 Kim Jihie 2 jihie.kim@dgu.edu
1 Department of Artificial Intelligence, Dongguk University , Seoul , Republic of South Korea
2 Department of Computer Science and Artificial Intelligence, Dongguk University , Seoul , Republic of South Korea
Coelho Paulo Jorge
Electronic publication date: 2025 May 2
Publication date: 2025
Volume: 11
Electronic Location ID: e2843
Received 2024 Aug 23; Accepted 2025 Mar 27
Copyright: © 2025 Lee and Kim
Copyright year: 2025
Copyright holder: Lee and Kim
License: This is an open access article distributed under the terms of the Creative Commons Attribution License, which permits unrestricted use, distribution, reproduction and adaptation in any medium and for any purpose provided that it is properly attributed. For attribution, the original author(s), title, publication source (PeerJ Computer Science) and either DOI or URL of the article must be cited.
License URL: https://creativecommons.org/licenses/by/4.0/

Keywords: Human pose estimation, Estimation, Segmentation, Restoration, Amodal instance segmentation

Funding: MSIT (Ministry of Science and ICT), Korea ITRC (Information Technology Research Center) IITP-2025-2020-0-01789 Artificial Intelligence Convergence Innovation Human Resources Development IITP-2025-RS-2023-00254592 IITP (Institute for Information & Communications Technology Planning & Evaluation) This research was supported by the MSIT (Ministry of Science and ICT), Korea, under the ITRC (Information Technology Research Center) support program (IITP-2025-2020-0-01789), and the Artificial Intelligence Convergence Innovation Human Resources Development (IITP-2025-RS-2023-00254592) supervised by the IITP (Institute for Information & Communications Technology Planning & Evaluation). The funders had no role in study design, data collection and analysis, decision to publish, or preparation of the manuscript.

==============================
Human pose estimation (HPE) is a field focused on estimating human poses by detecting key points in images. HPE includes methods like top-down and bottom-up approaches. The top-down approach uses a two-stage process, first locating and then detecting key points on humans with bounding boxes, whereas the bottom-up approach directly detects individual key points and integrates them to estimate the overall pose. In this article, we address the problem of bounding box detection inaccuracies in certain situations using the top-down method. The detected bounding boxes, which serve as input for the model, impact the accuracy of pose estimation. Occlusions occur when a part of the target’s body is obscured by a person or object and hinder the model’s ability to detect complete bounding boxes. Consequently, the model produces bounding boxes that do not recognize occluded parts, resulting in their exclusion from the input used by the HPE model. To mitigate this issue, we introduce the Restoring Occluded Mask Image for 2D Human Pose Estimation (ROM-Pose), comprising a restoration model and an HPE model. The restoration model is designed to delineate the boundary between the target’s grayscale mask (occluded image) and the blocker’s grayscale mask (occludee image) using the specially created Whole Common Objects in Context (COCO) dataset. Upon identifying the boundary, the restoration model restores the occluded image. This restored image is subsequently overlaid onto the RGB image for use in the HPE model. By integrating occluded parts’ information into the input, the bounding box includes these areas during detection, thus enhancing the HPE model’s ability to recognize them. ROM-Pose achieved a 1.6% improvement in average precision (AP) compared to the baseline.

Introduction

2D human pose estimation (HPE) is a research field that focuses on estimating human poses by detecting key points within a given image. HPE uses either a top-down or bottom-up approach. The top-down approach includes two stages: (1) detecting the target and drawing a bounding box around it, and (2) cropping the bounding box to use as input for the HPE model. This approach is faster and simpler than the bottom-up approach, but is heavily based on the accuracy of the detected bounding box. If the target is not detected in the first stage, it is excluded from the input for the second stage, leading to a failure in pose estimation. Thus, the importance of accurate detection of the boundary box is emphasized in the top-down approach. In contrast, the bottom-up approach detects all human joint coordinates in a single stage, grouping them into human poses. Although this approach is more accurate, it requires more computational time for the estimation. In this study, we employ the top-down approach for pose estimation.

The aim of HPE is to estimate the human pose, comprised of 17 key point coordinate pairs (nose, eyes, shoulders, elbows, wrists, hip, knees, ankles) as defined in the Common Objects in Context (COCO) dataset by Lin et al. (2014). Estimating key points for extremities, such as wrists and ankles, is challenging, while areas like the face (nose, eyes), shoulders, and hip are easier to detect, even when partially occluded. However, in images where a person’s hands, feet, wrists, or ankles are covered by other objects (e.g., a bicycle), these parts are difficult to detect, making HPE challenging under occlusion. These situations are referred to as occlusion, highlighting the challenge in ensuring that key points in occluded parts of the image are accurately identified.

Handling occlusion in HPE has always been challenging. Recent developments, such as the realistic augmentation approach by Ansarian & Amer (2021), tackle this issue by enhancing existing datasets with occlusion scenarios, thus making models more robust under real-world occluded conditions. This method introduces realistic occlusions into datasets like Human3.6 M, accurately reflecting natural occlusion phenomena and enabling model improvement without requiring architectural changes. Similarly, other approaches have investigated various strategies to manage occlusion, including the deployment of generative models like GAN, (Goodfellow et al., 2020), to reconstruct missing parts. However, these solutions frequently rely on artificial datasets or models with inherent biases, diminishing their effectiveness in real-world applications where occlusion varies unpredictably.

Building on this research, we propose a method that employs restored grayscale mask images to improve bounding box accuracy and, consequently, the accuracy of pose estimation. Our proposed method is illustrated in Fig. 1. We utilize the Whole COCO dataset, which comprises both visible and occluded mask annotations generated automatically from existing keypoint coordinates. This data set allows for more efficient training compared to manual annotation approaches, such as those used in COCOA (Zhu et al., 2017), which require substantial human labor. Furthermore, by superimposing these restored grayscale masks onto RGB images, we circumvent occlusion limitations inherent in the top-down approach, ensuring the inclusion of both visible and occluded body parts in bounding box detection.

Figure 1 (A) ROM-Pose structure. (B) The restoration model uses faster R-CNN to generate an occluded mask image and an occludee mask image from the RGB image. These two generated images are compared to delineate the boundary line between them. Based on the identified boundary line, the occluded mask image is restored by utilizing numpy to fill obscured areas with a value of 1. Then, the restored occluded mask image is overlaid onto the RGB image. This combined image, depicted in (C), serves as an input for the HPE model. The HPE model then estimates 17 key points. However, if certain key points are not detected, these undetected key points are typically located near the corners of the image. The difference in the input for the HPE model between the baseline and ROM-Pose is demonstrated in (D) and (E).

Image source: COCO Dataset © COCO Consortium, CC BY 4.0.

Our approach builds on the understanding that realistic occlusion scenarios, as demonstrated by Ansarian & Amer (2021), and precise boundary restoration using Generative Adversarial Network (GAN), as suggested by Goodfellow et al. (2020), are essential for enhancing human pose estimation performance in occluded conditions. In this study, we aim to improve model performance in environments with occlusion by deploying restored grayscale mask images to maintain occluded parts during bounding box detection (Jiang et al., 2024).

In summary, this study contributes significantly by (1) introducing the Whole COCO dataset, an automatically generated dataset featuring both visible and occluded mask annotations, substantially reducing the necessity for manual labor and time-consuming annotation processes; and (2) proposing a method that uses restored grayscale mask images, which ensures the inclusion of occluded body parts in bounding box detection, thereby enhancing pose estimation accuracy by 1.6% over the baseline. These innovations underscore the significance of realistic occlusion handling in 2D pose estimation and illuminate the potential for enhanced model generalization in real-world settings.

Related work

Human pose estimation

HPE is a critical task in computer vision, with various methods developed to detect and associate body parts. These methods are typically classified into two broad categories: top-down and bottom-up approaches, each exhibiting unique strengths and weaknesses concerning speed, scalability, and accuracy.

The top-down approach, as described by Huang, Gong & Tao (2017), Papandreou et al. (2017), Sun et al. (2019a), Wang et al. (2020), initially detects individuals in an image before applying a bounding box for pose estimation. This method provides beneficial scalability but heavily relies on the precision of the bounding box. To increase precision, (Fang et al., 2017) resolved challenges in pose estimation, including those with imprecise bounding boxes, and (Khirodkar et al., 2021) suggested deploying multiple bounding boxes during non-maximum suppression (NMS) to better detect poses, particularly in occluded subjects. This top-down approach aligns well with the methodologies advocated in this article as it relies on bounding box-based human pose estimation.

In contrast, the bottom-up approach, as discussed by Pishchulin et al. (2016), Insafutdinov et al. (2016), Cao et al. (2017), Kreiss, Bertoni & Alahi (2019), Cheng et al. (2020), Geng et al. (2021), and Jin et al. (2020), focuses on detecting key points for body parts and grouping them to estimate the full pose. Although highly accurate, this method is computationally intensive because it entails detecting all key points before grouping them. Efforts like those by Debapriya et al. (2022) and PersonNet by Papandreou et al. (2018) have enhanced efficiency through multilayer feature extraction and the application of 2D offset fields to facilitate key point grouping. However, bottom-up methods demand extensive computation and are unsuitable for the applications proposed in this study.

HPE involves estimating the pose of humans in an image. It faces particular challenges in detecting peripheral body parts like hands and feet, which are often smaller and more easily occluded than other body parts such as the head or torso. To enhance accuracy in HPE, various backbone models with improved architectures have been proposed.

For example, Newell, Yung & Deng (2016) introduced stacked hourglass networks to address resolution discrepancies stemming from the varying scale of detected individuals. By transforming input images into multiple scales, the model aggregates results from each to improve pose estimation accuracy. Sun et al. (2019b) developed HRNet, which maintains high resolution for feature extraction, regardless of person sizes in the image. Additionally, Gao et al. (2019) proposed Res2Net, a residual connection-based structure to deepen networks and enhance feature extraction, which serves as the foundation for the simple baseline, while Su et al. (2019) introduced the channel shuffle module (CSM) to improve cross-channel information sharing between low and high-level feature maps.

Another breakthrough in HPE is the introduction of Vision Transformers (ViT) by Dosovitskiy et al. (2021), which process images using a decoder-encoder structure. This model, as seen in models such as Yufei et al. (2022), outperforms traditional convolutional neural networks (CNN) models in feature extraction. Recently, Zhou et al. (2023) have developed hybrid models that combine bottom-up detectors with attention-based top-down methods (BUCTD) to integrate the strengths of both approaches.

Amodal instance segmentation

Amodal instance segmentation as described in Li & Malik (2016) aims to restore occluded parts of objects in an image, enabling the model to deduce hidden parts based on available visual cues. This approach mirrors human cognitive capabilities, which allow individuals to perceive an entire object even when parts are concealed. Various studies, such as those by Zhan et al. (2020), have explored amodal instance segmentation through self-supervised learning to anticipate occluded object sections. Qi et al. (2019) introduced the KITTI (Geiger, Lenz & Urtasun, 2012) INStance dataset (KINS), augmenting it with detailed pixel-level annotations for visible and occluded object regions, enabling advancements in amodal instance segmentation. Ke, Tai & Tang (2021) uses a bilayer structure to decouple occluder and occludee boundaries, improving segmentation, especially in heavy occlusion cases. Nonetheless, these methodologies are often challenged by lengthy restoration periods and substantial computational demands.

To mitigate these challenges, Nguyen & Todorovic (2021) introduced a technique that utilizes uncertainty loss to accelerate restoration, concentrating on boundary information and thus diminishing the computational burden. Conversely, our study automates dataset generation, obviating the need for manual annotation and optimizing the training regimen for amodal instance segmentation, making the process more rapid and efficient.

In our work, we propose a novel method that combines a restored mask with an RGB image to enhance pose estimation accuracy. This method enables more precise detection of key points, even in the presence of occluded parts and inaccurate bounding boxes, setting it apart from traditional approaches in HPE.

Proposed method

ROM-Pose consists of two main components: a restoration model and a pose estimation model (HPE model). The restoration model restores obscured parts of the target, while the HPE model derives poses from the restored images. The process involves three key steps: (1) dataset creation for training the restoration model; (2) training the restoration model; and (3) training the pose estimation model.

Whole COCO mask dataset

To restore the mask image, it is essential to train the restoration model using both full-mask and occluded images. Full-mask images are grayscale mask images that encompass both occluded and visible parts of the target. Since the COCO 2017 dataset lacks these full-mask images, we propose a method to generate them in order to effectively train the restoration model.

An annotation file from the COCO dataset, including image ID, key-point coordinates, and mask image coordinates, is needed to produce full-mask images. The existing mask images in the COCO dataset are inadequate because they fail to cover occluded areas. For instance, Fig. 2C displays a gap in the COCO mask within a red circle, with ground truth key points (pink dots) present, underscoring the necessity for full-mask images to enable the restoration model to recover missing parts.

Figure 2 Dataset for restoration model training.

(A) The original RGB image sourced from the COCO dataset. (B) Key point coordinates (pink dots) overlaid onto the RGB image. These are key points that the HPE model ultimately estimates. (C) A grayscale mask image depicting visible parts of the target in the RGB image. Occluded key points are marked as red circles. (D) Hidden key points as projected (C) onto (A). (E) An image created based on key points. (F) Process of combining (C) and (E). (G) The created ground truth image. (H) Overlay of (G) image onto (A). Image source: COCO Dataset © COCO Consortium, CC BY 4.0.

Furthermore, experiments conducted using the initial mask images from the original COCO dataset revealed that there was no significant difference in performance between models trained with these masks and those trained without masks.

To solve this issue, we suggest an automated method for creating full-mask images, which drastically cuts down on the time and labor associated with manual annotation. However, the effectiveness of this method is constrained by the accuracy and completeness of the existing key-point annotations, which may limit its general applicability.

Our full-mask generation method is applicable not only to the COCO dataset but also to other COCO-style formatted datasets like OCHuman (Zhang et al., 2019), which includes human key-point pairs. This wide applicability demonstrates the method’s potential value in occlusion-aware human pose estimation across various datasets.

The process for generating the key points image starts by loading the original image onto an empty canvas. A circle is then drawn at each key point’s coordinates, as illustrated in Fig. 3A. Subsequently, ellipses are configured based on key-point relationships, using the midpoint as the origin. In a deviation from simply drawing circles, ellipses connect coordinates representing body parts to create a complete mask, as depicted in Fig. 3B.

Figure 3 Key points image generation.

(A) The original ground truth images from the COCO dataset, marked with green dots to indicate human body key points, are set against a black background, matching the size of the RGB images in the dataset. (B) An illustration of ellipses drawn around the center of the ground truth points, with the combined result of these ellipses shown. (C) An incorrectly combined image where the generated key points image obscures the underlying COCO mask, thereby obstructing crucial details of the mask. (D) The appropriate technique for creating the key points image involves drawing ellipses around the key points using the midpoint between two coordinates that represent human body parts. (E) The final key points image effectively integrates the COCO mask and the newly generated key points image, maintaining the detailed contours of the COCO mask while filling in the gaps necessary for accurate ground truth detection.

If the mask were created strictly following the ground truth, as demonstrated in Fig. 3B, fine lines from the original COCO dataset would be covered by the overlaid key points image, resulting in the visibility of white areas. Nonetheless, as evidenced in Fig. 3C, our method retains the integrity of the original COCO dataset’s mask.

Given that this technique is designed to enhance the original COCO dataset, it is not intended to replace it directly. Consequently, we suggest producing the key points image as depicted in Fig. 3D, identifying human body key points and surrounding them with ellipses. The red and blue ellipses and lines in Fig. 3D demonstrate this technique. In Fig. 3E, the created key points image enhances the detailed mask of the COCO dataset, supporting more precise ground truth identification. While shapes like triangles or rectangles can also be used to represent human pose, we found that ellipses are more effective in capturing the natural relationships between key points. Therefore, in this study, we use ellipses for mask generation. However, depending on the dataset being applied, using rectangles or other shapes for mask generation is also feasible, providing flexibility in dataset creation.

Ellipses are drawn using Eqs. (1) and (2). In Eq. (1), Rfactor modifies the dimensions of circles and lines in proportion to the bounding box’s size, where w represents the width and h the height. Equation (2) computes the circle’s radius used for displaying key points, ensuring it ranges between 4 and Rfactor×10.

(1) Rfactor=(wmax−wmin)×(hmax−hmin)imageheight×imagewidth

(2) Rpoint=min(10,max(Rfactor×10,4))

(3) Full-mask=Maskimage⊙0.5+keypointimage⊙0.5.

The final key points image is combined with the mask image through element-wise multiplication and summation, as defined in Eq. (3), to produce a full-mask image. This procedure blends the mask and key points images with equal weight, achieving a semi-transparent appearance. These full-mask images are produced for each image in the COCO 2017 dataset, creating the comprehensive Whole COCO mask dataset.

Restoration model training

We trained the restoration model using the Whole COCO mask dataset. Figure 4A depicts an object obstructing the target. Figure 4C displays the grayscale mask image of the obstructing object, termed the occludee mask image. Figure 4C also shows the target area that needs restoration. Figure 4B illustrates the area where the restoration model identifies the target in the image. At this point, the selection of the occludee mask from the dataset pair is essential. For instance, the first line of Fig. 4 presents two men in the picture, limiting the maximum mask information obtainable from the image to 2. Therefore, if the individual at the back is the target, the occludee must be the person in front in the image. In scenarios with more than two individuals, like in the 2nd or 3rd row of Fig. 4, the nearest person is selected as the occludee mask. After selecting the occludee target and the occluded target, the model is trained to determine the boundary between the occluded mask and the occludee mask. Figure 4D demonstrates the boundary line as identified by the model. The dataset does not include ground truth data for the boundary line.

Figure 4 Learning process of the restoration model.

(A) displays an obstacle overlaid onto the original image. (B) represents a visible grayscale mask image corresponding to the part estimated to be the target in the RGB image. (C) depicts an obstacle grayscale mask image, referred to as the occludee. (D) illustrates the contour (or edge) between (B) and (C). (E) shows the complete full-mask image of the target (ground truth). (F) demonstrates the result of the restoration model applied to (B). Image source: OCHuman Dataset © 2018 Roy Tseng, licensed under the MIT License.

To achieve this, we leveraged the Amodal Segmentation with Boundary Uncertainty (ASBU) (Nguyen & Todorovic, 2021) method. The ASBU method extends UNet (Ronneberger, Fischer & Brox, 2015) to output an H×W×2 feature map with two channels: one for

(4) L=∑i=1N1ui⊮(mi∗=0)Li,0+λ⊮(mi∗=1)Li,1,

In Eq. (4), the following symbols are used: ui is a weighting factor for the i-th pixel, mi∗ is the ground truth mask for the i-th pixel, where mi∗=0 indicates the pixel is outside the mask and mi∗=1 indicates the pixel is inside the mask, Li,0 is the loss for the i-th pixel when it is outside the mask, Li,1 is the loss for the i-th pixel when it is inside the mask, and λ is a regularization factor applied to the loss for pixels inside the mask.

Figure 4E represents the ground truth mask that the model aims to predict, while Fig. 4F displays the reconstructed mask generated by the model. The ground truth masks in Fig. 4E were created using the newly proposed COCO whole dataset, an extension of the original COCO dataset developed specifically for this study. Please refer to the description in Fig. 3 for more details on the COCO whole dataset and the mask generation process.

To evaluate the model’s performance, we compare the area covered by the predicted mask in Fig. 4F with the area of the ground truth mask in Fig. 4E, calculating the loss using two specific functions. Loss1 (Eq. (5)) assesses reconstruction error within the target object by focusing on regions where mask =1 and applies inmask_weight to emphasize restoration within the object’s boundaries. On the other hand, Loss2 (Eq. (6)) addresses error outside the target in regions where mask =0, using outmask_weight to avoid unnecessary reconstructions. Through iterative refinement, the model minimizes discrepancies between the reconstructed and ground truth masks (Figs. 4F and 4E).

Figure 4 also includes intermediate steps such as the occluder (Fig. 4C) and the overlap region (Fig. 4D), which assist the model in boundary detection and accurate restoration.

(5) Lin=λin1|M|∑i∈M12(yi−μσ)2

(6) Lout=λout1|B|∑i∈B12(yi−μσ)2.

Here, λin is a weighting factor applied to the loss computed for pixels inside the mask region M, where M denotes the set of indices corresponding to the masked region ( mi=1). This ensures that the model prioritizes accurately reconstructing occluded regions. Conversely, λout is applied to the loss for pixels in the background region B, where B denotes the set of indices corresponding to the background region ( mi=0), aiding the model in differentiating the target from the background. The mean squared error (MSE) is employed to measure the overall discrepancy between the predicted mask and the target full-mask image.

The model iteratively refines the restored mask image until it closely resembles the full-mask image. During testing, the restoration model no longer requires the full-mask image, as it has gained sufficient confidence in identifying boundary lines. Through these adaptations, our method utilizes the ASBU framework while integrating key point information and dataset-specific adjustments to improve the accuracy and reliability of mask restoration in occluded scenarios.

Pose estimation with restored image

To estimate human pose, the pose estimation model first detects the bounding box around the target object within the input image. This image is then enhanced with a restored grayscale mask, generated by our restoration model, to capture occluded regions. Within each identified bounding box, a 64×48 heatmap is generated for each joint, and the coordinates of each estimated joint are derived from these heatmaps.

For pose estimation, we utilized the Simple Baseline method, which employs ResNet (He et al., 2016), a commonly used backbone network for image feature extraction. This method adds several deconvolutional layers above ResNet’s final convolutional layer, known as C5, to produce heatmaps from deep, low-resolution features. This simple design closely aligns with state-of-the-art methods like Mask R-CNN (He et al., 2017).

The Simple Baseline model, by default, comprises three deconvolutional layers equipped with batch normalization (Ioffe & Szegedy, 2015) and ReLU activation (Nair & Hinton, 2010). Each layer utilizes 256 filters with a 4×4 kernel and a stride of 2, succeeded by a 1×1 convolutional layer that generates the predicted heatmaps, {H1,…,Hk}, for all k key points. This efficient architecture eliminates the complexity of skip connections found in other models.

Our method enhances the Simple Baseline approach by incorporating the newly developed Whole COCO dataset, an improved version of the COCO dataset that features both original RGB images and restored grayscale mask images. This integrated dataset enables the model to learn from both visible and occluded key points, thereby improving its pose estimation capabilities under conditions of occlusion.

To develop the Whole COCO dataset, we processed original COCO images to create restored grayscale mask images using our restoration model, and then merged them with the RGB images. The pose estimation model underwent training on these combined images using standard augmentation techniques to boost robustness and prevent overfitting. We assessed the model’s performance on a test set, focusing on accuracy in pose estimation under various occlusion conditions. Evaluation metrics included average precision (AP) and average recall (AR).

We employed MSE as the loss function. A metric extensively used in machine learning, MSE measures the mean squared difference between predicted and actual values. In pose estimation, MSE quantifies the discrepancy between the coordinates predicted by the model ( Y^i) and the actual ground truth coordinates ( Yi) for each of the 17 joints considered, with the average MSE across all joints serving as the loss metric. The targeted heatmap H^k for each joint k is formed by applying a 2D Gaussian centered at the ground truth location of the k-th joint.

Finally, the pose estimation process involves both the restoration and pose estimation models, as illustrated in Fig. 1. The occluded mask and occludee mask images are extracted from the original RGB image via Faster R-CNN and then restored by the restoration model. The restored mask image and the RGB image are subsequently input into the pose estimation model. The restored mask image is overlaid onto the RGB image at full opacity and scale. Without the restored grayscale mask, the model cannot recognize occluded regions as part of the target. However, with the restored grayscale mask, the model can accurately define a bounding box that includes occluded regions, enabling it to estimate hidden key points of the target. This enhancement substantially improves pose estimation accuracy.

Experiments

Experimental setup

Datasets

The dataset of COCO from Lin et al. (2014) is notable as one of the most extensive datasets created by Microsoft. It is widely used across various computer vision domains, including image recognition, object detection, segmentation, and key points detection. Comprising 80 diverse object categories, each image in the COCO dataset includes multiple instances of objects. Detailed annotation information accompanies each image, which provides bounding box coordinates delineating object locations, mask data for object segmentation, and key points coordinates for object analysis. In this study, we leveraged the COCO dataset, encompassing images along with their associated key points and mask data. Specifically, we focused on images featuring human subjects, selecting them for analysis from within the COCO dataset. To achieve this, we utilized a dedicated annotation file containing information pertinent to human-containing images.

The OCHuman dataset is derived from the COCO dataset but is smaller in scale. It notably contains a higher frequency of occluded images compared to COCO. It is frequently employed as a test set in pose estimation research, with COCO typically serving as the training dataset. Similar to the COCO dataset, OCHuman provides annotation files with key point coordinates for 17 key points and mask information. In this article, we emphasize the significance of including essential information from occluded regions for accurate pose estimation. Hence, we utilized this dataset. Training was conducted using the COCO dataset, and testing was performed using the OCHuman dataset.

Evaluation metric

AP and AR were used as evaluation metrics in this study. AP served as an indicator of the accuracy of the object detection model, calculated as the average area under the precision-recall curve. Precision denotes the ratio of correctly identified objects to those detected by the model, whereas recall indicates the ratio of actual objects correctly identified. AP values, which range from 0 to 1, with higher values indicating better model performance, were also computed at an Intersection over Union (IoU) threshold of 50% (AP50), a commonly used threshold in object detection. AP50 represents the average precision for cases where the IoU between an object and the predicted bounding box exceeds 0.5, highlighting its importance in measuring model accuracy. AP75 indicates cases where the IoU is 0.75 or higher. AP-medium (AP-M) and AP-large (AP-L) calculate AP for medium-sized and large-sized objects, respectively. AR, representing the average of recall values, was computed by averaging recall values across various thresholds, making it a valuable metric for assessing the model’s ability to effectively detect various poses. AR-medium (AR-m) evaluates recall for medium-sized objects.

Implementation details

In this article, our core model was ResNet from He et al. (2016), pretrained on ImageNet from Krizhevsky, Sutskever & Hinton (2012), selected for its robust feature extraction capabilities. Experiments involved several parameters, including ResNet50, ResNet101, and ResNet152 architectures, with an input image resolution of 256 × 192 pixels. Training proceeded over 140 epochs with a batch size of 64, conducted in a setup equipped with four NVIDIA A5000 GPUs leveraging the PyTorch framework. The training concluded after 54,000 iterations, dictated by convergence observed in mask restoration performance.

Experiment result

As shown in Table 1, AP increased by 1.6% with ResNet-50, by 1.6% with ResNet-101, and by 1.8% with ResNet-152. Although the improvement in AP may seem modest, other metrics, particularly AP-medium (AP-m), demonstrated significant enhancements, increasing by over 10%. This result underscores the model’s ability to effectively utilize overlaid mask information for medium-sized objects, better capturing occluded parts in the restored mask and thus aiding in accurate posture estimation. For small-sized objects, the mask’s impact was minimal due to their lower resolution and smaller dimensions.

Table 1 Comparison of model performance on the COCO validation set shows training on the COCO train set using an input resolution of 256 × 192.

Architectures compared include SBL (baseline) and Rom-Pose (Ours), which utilize ResNet-50, ResNet-101, and ResNet-152. The article also specifies the number of parameters for each architecture. Furthermore, ViTPose results are presented for juxtaposition, highlighting differences in performance and computational efficiency, albeit only for specific metrics (AP and AR).

Method	Architecture	#Params	AP	AP 75	AP-M	AP-L	AR	AR-m	
SBL (baseline)	ResNet-50	34.0 M	70.5	78.0	68.2	75.9	73.6	71.1	
Rom-Pose (Ours)	ResNet-50	34.0 M	72.1	76.4	80.8	61.1	75.4	83.0	
			(+1.6)		(+12.6)			(+11.9)	
SBL (baseline)	ResNet-101	53.0 M	71.9	79.3	69.6	76.7	75.5	72.3	
Rom-Pose (Ours)	ResNet-101	53.0 M	73.5	80.3	81.9	62.5	75.8	83.9	
			(+1.6)		(+12.3)			(+11.6)	
SBL (baseline)	ResNet-152	68.6 M	72.1	79.8	70.4	77.2	75.9	73.0	
Rom-Pose (Ours)	ResNet-152	68.6 M	73.9	81.2	82.9	62.5	76.5	84.8	
			(+1.8)		(+12.5)			(+11.8)	
ViTPose-B	Vit-Base	86.0 M	75.8	–	–	–	81.1	–	
ViTPose-H	Vit-Huge	632.0 M	78.1	–	–	–	84.1	–	

However, for large-sized objects, applying a mask could occasionally obstruct feature extraction. The mask’s increased size, based on the object’s dimensions, results in an expanded masking area, similar to the effect shown in Fig. 3C. In such cases, the mask image may obscure essential features of the target object, potentially reducing the accuracy of posture estimation.

Compared to state-of-the-art models like ViTPose (Yufei et al., 2022), our model exhibits a modest decrease in AP performance. However, our method provides significant computational efficiency benefits, as it has considerably fewer parameters. Moreover, our model adopts a top-down approach, which simplifies the pipeline by separating person detection and key point estimation. This modularity not only enhances interpretability but also allows for flexible integration into existing systems. As shown in Table 1, this reduced parameter count means our model does not require high-spec hardware, thus it is more accessible and practical for deployment in resource-constrained environments compared to Vision Transformer-based models. Despite these specifications, the model shows significant improvements in certain evaluation metrics, such as AP-m, suggesting it can achieve competitive performance at much lower computational costs.

Figure 5 is a qualitative result of performance improvement, where Fig. 5A visualizes the correct ground truth locations of 17 key points in the OCHuman dataset. Figure 5B displays results using the existing baseline, while Fig. 5C illustrates 17 key points identified by ROM-Pose.

Figure 5 Presents the qualitative results, demonstrating the performance improvement of the proposed model.

(A) Ground truth key points. (B) Baseline model result. (C) Our model result, focusing on colored boxes in the image. Image source: OCHuman Dataset © 2018 Roy Tseng, licensed under the MIT License.

Comparing the color boxes of each image, despite the unchanged model size, the key points predicted by ROM-Pose were closer to the correct ground truth than those predicted by the baseline model. Several observations can be made from the color boxes of each image. In the first image on the left, where two individuals overlapped, attention shifts to the second yellow box. With the baseline model, joints overlapped, as indicated by the presence of white and blue dots within the same box. However, ROM-Pose accurately identified this overlap as a single joint, aligning closely with the correct depiction shown in Fig. 5A. In the second image from the left, the positioning of the man’s arm (blue dot) within the orange box was closer to the correct answer. In the third image from the left, the focus went to the green box, where the key point (white point) on the lower body of the individual in red attire was more accurately captured by ROM-Pose, aligning well with the correct answer. In the fourth image from the left, attention was drawn to the red box marking the foot, which was correctly recognized as belonging to a single individual rather than two separate people’s feet. Regarding the sixth image from the left, the number of blue dots within the blue box increased compared to the baseline model, indicating that the model accurately identified the presence of two joints within the box. This aligned with the correct answer and confirmed that ROM-Pose demonstrated greater robustness than the baseline model in estimating posture in occluded environments.

Ablation study

To assess the impact of varying epsilon values (the transparency ratio between the grayscale mask and the RGB image) on model performance, a detailed ablation study was conducted. Specifically, the effects of different epsilon values, including small values such as 1e−7 and 1e−5, were examined to evaluate their influence on training. Additionally, when the epsilon value exceeds 0.1, the model achieves an accuracy above 85%. However, this results in excessive reliance on the mask image, leading to an overfitted model. Therefore, in this ablation study, we set the maximum epsilon value to 0.1 to ensure balanced model generalization. Furthermore, additional experiments were performed to investigate whether the mask image significantly influences the training process. This study demonstrates that incorporating the mask image as part of the input enhances the model’s ability to estimate human pose. By varying the intensity levels of the mask image, we analyzed its direct contribution to the model’s capacity to effectively capture structural information.

(7) input=(rgbimage⊙(1−ϵ)+maskimage⊙ϵ).

The epsilon values were adjusted as specified in Eq. (7) and the input was subsequently normalized to the range of [0, 255] to ensure consistent model processing. The findings summarized by different epsilon values are recorded below: Epsilon = 0.000001: The results (denoted by ⋆ in Table 2) showed no noticeable improvement over the baseline. With such a small epsilon value, the mask image had minimal effect, and the model’s performance was similar to that of the baseline.

Epsilon = 0.00001: The results (denoted by † in Table 2) demonstrated a significant enhancement in model accuracy. This increase in transparency enabled the model to integrate more features from the occluded portions of the object, consequently improving performance.

Epsilon = 0.0001: Results (indicated by ∙ in Table 2) showed a moderate improvement in performance compared to epsilon = 0.00001. Although AP did not significantly improve, AR increased, illustrating the model’s enhanced capability to estimate a wider range of poses by utilizing occluded parts.

Epsilon = 0.001: Experimental results (indicated by ▴ in Table 2) displayed no significant enhancement with ResNet50. However, significant improvements were observed with ResNet101 and ResNet152, suggesting that deeper ResNet architectures are more effective at exploiting mask image features.

Epsilon = 0.1: Experimental results marked with ♢ in Table 2 and illustrated in Fig. 6, optimal performance was achieved when the mask and RGB images were equally weighted. An example of the most effective input image is displayed in Fig. 3E. Higher epsilon values resulted in enhancements in pose estimation, confirming the effectiveness of superimposing the grayscale mask image.

Figure 6 AP results based on different epsilon values in comparison with the baseline model.

Table 2 Accuracy of pose estimation model according to the ratio of overlapping masks ( ⋆, †, ∙, ▴ and ♢ denote ϵ of 0.000001, 0.00001, 0.0001, 0.001 and 0.1, respectively).

Method	AP	AP50	AP75	AR	
SBL (baseline) 50	70.5	91.5	78.0	73.6	
SBL (baseline) 101	71.9	91.5	79.3	75.5	
SBL (baseline) 152	72.1	91.2	79.8	75.9	
ROM-Pose 50 ⋆	70.5	91.4	78.1	73.7	
ROM-Pose 101 ⋆	71.9	91.5	79.3	76.1	
ROM-Pose 152 ⋆	72.1	91.2	79.7	75.9	
ROM-Pose 50 †	71.1	91.5	78.3	74.2	
ROM-Pose 101 †	71.9	91.6	79.3	76.0	
ROM-Pose 152 †	72.2	91.3	80.0	76.0	
ROM-Pose 50 ∙	71.1	91.5	78.3	74.7	
ROM-Pose 101 ∙	72.0	92.0	79.8	76.0	
ROM-Pose 152 ∙	72.8	92.3	80.4	75.8	
ROM-Pose 50 ▴	71.2	91.5	79.2	74.9	
ROM-Pose 101 ▴	72.4	92.3	80.1	76.0	
ROM-Pose 152 ▴	73.2	92.4	81.1	76.3	
ROM-Pose 50 ♢	72.1	89.5	76.4	75.4	
ROM-Pose 101 ♢	73.5	92.5	80.3	75.8	
ROM-Pose 152 ♢	73.9	89.6	81.2	76.5	

Discussion

This study highlights the potential of employing grayscale mask images to improve human pose estimation, particularly by using the Whole COCO dataset. The model exhibited notable improvements in occlusion scenarios, facilitating more accurate pose estimation, particularly for medium-sized objects (AP-M). The expanded dataset, which includes both visible and occluded regions, contributed to improved generalization in real-world images.

One of the key findings is the notable improvement in AP-M, which assesses the model’s performance for medium-sized objects. This result indicates that the model effectively utilizes the overlaid mask information for these objects, where occluded parts are better represented in the restored mask, thus aiding in accurate pose estimation. However, the overall AP did not demonstrate significant improvements across all object sizes. Specifically, the AP-L, AP50 and AP75 metrics slightly declined. This decline can likely be attributed to the increased mask size obscuring some context necessary for precise pose estimation in larger or more clearly visible objects, where the occlusion effect is less pronounced. Future work should focus on optimizing the mask size and its application across various object sizes to mitigate these effects.

The improvement in AP-M is notable; however, the absence of substantial enhancement in other metrics such as AP-L and AP50 necessitates further investigation. Specifically, the method may better accommodate medium-sized objects because of their balance between resolution and occlusion, making them particularly receptive to the proposed restoration. However, larger objects may experience detrimental effects due to the excessive application of the mask, resulting in the occlusion of critical areas of the target object. This could explain the minor improvements or even a decrease in performance for large objects.

Although the model excelled with medium-sized objects, its difficulties with larger and smaller objects highlight the need for further refinement. Moreover, the integration of the restoration and pose estimation models increased computational complexity, posing a significant challenge for real-time applications. This balance between accuracy and computational efficiency must be prioritized in future research, especially for practical implementation.

Furthermore, applying super-resolution techniques to enhance the quality of the restored masks for smaller objects could benefit the model’s efficacy across different object sizes. Also, adjusting the mask size in relation to the object may enhance performance at varying scales and levels of occlusion.

In addition, through our ablation study, we found that the transparency setting of the restored grayscale mask images significantly impacts the model’s performance. The best performance was achieved when the transparency was set to match that of the RGB image. This result provides direction for future model optimization.

Currently, we employ simple geometric shapes to generate grayscale mask images as part of our automation efforts. This approach was selected to simplify data generation and facilitate understanding through a straightforward methodology.

In future research, we plan to analyze the regions on which the model focuses when estimating poses. By utilizing backpropagation, we aim to identify these areas and define boundaries and specific regions. For instance, we intend to expand the boundary areas identified by the model (e.g., by 2 cm) and use these expanded regions to generate new grayscale masks, replacing the current ones. Future proposals do not require the creation of any additional datasets.

In conclusion, while ROM-Pose markedly enhances pose estimation for medium-sized objects, addressing its shortcomings, particularly in balancing accuracy with computational complexity, and improving its generalization across diverse object sizes and occlusion levels, remains crucial. Future endeavors should concentrate on refining mask generation, incorporating high-resolution data, and exploring advanced techniques such as super-resolution to boost the model’s robustness and precision.

Conclusion

Occlusions in images present significant obstacles to the accuracy of human pose estimation models. This study tackles these challenges through ROM-Pose, a methodology devised to restore occluded regions and integrate them into the model input. Our findings reveal that occluded regions hold essential information that can augment pose estimation accuracy. ROM-Pose achieved enhancements in AP-M, emphasizing its capacity to utilize information from occluded areas effectively. Nonetheless, it did not surpass the baseline in AP-L, indicating that while ROM-Pose is adept at detecting medium-sized objects, it struggles with larger objects due to potential occlusion of vital contextual information.

Future work should focus on techniques that dynamically adjust mask sizes in relation to the target object, potentially through scalable image adjustments. Moreover, although ROM-Pose enhances accuracy, it may also increase computational complexity. Future research should aim to optimize the tradeoff between accuracy and complexity, especially in the context of real-time pose estimation. Addressing these challenges could further enhance the model’s ability to accurately estimate human poses across various occlusion conditions and object sizes.

Additionally, while ROM-Pose currently employs a two-stage approach, consisting of separate restoration and pose estimation models, future work could explore integrating these stages into a unified one-stage model. This could be achieved by utilizing feature maps for grayscale mask generation, simplifying the model and reducing its computational complexity, which would improve its efficiency for real-time applications.

Supplemental Information

Supplemental Information 1 ROM-Pose code.

Supplemental Information 2 Dataset making file for Whole COCO.

Additional Information and Declarations

Competing Interests

The authors declare that they have no competing interests.

Author Contributions

Yunju Lee conceived and designed the experiments, performed the experiments, analyzed the data, performed the computation work, prepared figures and/or tables, authored or reviewed drafts of the article, developed a novel data collection method specifically for this study. Provided critical theoretical frameworks that significantly guided the analysis and interpretation of the results, and approved the final draft.

Jihie Kim conceived and designed the experiments, authored or reviewed drafts of the article, and approved the final draft.

Data Availability

The following information was supplied regarding data availability:

The ROM-Pose model is available at GitHub and Zenodo:

- https://github.com/LEEYUN-JU/MaskPose.git.

- Allie. (2024). LEEYUN-JU/MaskPose: HPE (HPE). Zenodo. https://doi.org/10.5281/zenodo.14233079.

The COCO Dataset is available at https://cocodataset.org.

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
