# Peer review of "ROM-Pose: restoring occluded mask image for 2D human pose estimation"

_PeerJ Computer Science, doi:10.7717/peerj-cs.2843_

## Round 0.1 · original submission · Major Revisions

· Academic Editor

Major Revisions

Dear authors,

You are advised to critically respond to all comments point by point when preparing an updated version of the manuscript and while preparing for the rebuttal letter. Please address all comments/suggestions provided by reviewers, considering that these should be added to the new version of the manuscript.

Kind regards,
PCoelho

Reviewer 1 ·

Basic reporting

This manuscript 103542 a novel method called ROM-Pose to address the challenge of occluded human pose estimation in 2D images. The authors argue that occlusion significantly affects the accuracy of human pose estimation (HPE) models, especially when using the top-down approach. To mitigate this issue, ROM-Pose employs a restoration model that utilizes binary mask images to restore occluded parts of the target. This restored binary mask image is then overlaid onto the original RGB image, and the combined image is used as input for the HPE model. The authors introduce a new dataset called Whole COCO, which includes full-mask images for both occluded and visible parts, to train the restoration model. Experimental results demonstrate that ROM-Pose improves Average Precision (AP) by 1.6% compared to the baseline method, indicating better performance in handling occluded poses. It was a pleasure reviewing this work and I can recommend it for publication in PeerJ Computer Science after a major revision. I respectfully refer the authors to my comments below.

Experimental design

1. The English needs to be revised throughout. The authors should pay attention to the spelling and grammar throughout this work. I would only respectfully recommend that the authors perform this revision or seek the help of someone who can aid the authors.

2. (References) Please adjust the style of all the references to meet the PeerJ Computer Science requirement.

3. ROM-Pose's top-down approach outperforms bottom-up methods in occlusion scenarios by restoring the binary mask image of occluded areas, enabling more accurate keypoint detection. However, bottom-up methods face higher computational complexity due to detecting all keypoints.

4. The automated creation of the Whole COCO dataset reduces time and labor costs compared to manual annotation. This method's reliance on existing datasets' keypoint coordinates may limit its widespread application.

5. The introduction of restored binary mask images enhances ROM-Pose's robustness and generalization in occlusion situations. Yet, for small-sized targets, the restoration effect is limited by resolution issues, suggesting potential improvements through super-resolution techniques.

Validity of the findings

6. Ablation experiments reveal that the transparency of the restored binary mask images significantly impacts model performance, with optimal results when set equal to the RGB image. This insight provides direction for further model optimization.

7. While ROM-Pose improves accuracy, it may increase computational complexity compared to the baseline, particularly during binary mask image restoration and enhanced image generation. Future research should focus on maintaining accuracy while reducing complexity to improve real-time performance.

Additional comments

10. ROM-Pose's potential extends beyond human pose estimation, suggesting applicability in other computer vision tasks involving occlusion, such as vehicle detection and animal pose estimation. Future work could explore these applications.

11. The multi-stage model of ROM-Pose, consisting of a restoration model and a pose estimation model, may lead to error accumulation. Future research could investigate integrating these stages into a unified end-to-end model to reduce errors and enhance overall performance.

12. Discuss the pros and cons of the proposed human pose estimation models.

My overall impression of this manuscript is that it is in general well-organized. The work seems interesting and the technical contributions are solid. I would like to check the revised manuscript again.

Cite this review as

Reviewer 2 ·

Basic reporting

Clarity and Professionalism: The manuscript is generally written in clear English, but it appears that certain parts may have been influenced by translation software. As a result, several instances do not meet academic writing standards and still require correction. I recommend a thorough language review to enhance the professionalism and clarity of the text, ensuring it meets the conventions of scientific writing.

Introduction and Background: The introduction provides sufficient context to the study, including an overview of related literature, which is generally well referenced and relevant. However, there is a lack of a comprehensive discussion of occlusion-robust or occlusion-aware work, which should be an essential focus given that the study addresses the occlusion problem. Simply mentioning Human Pose Estimation in the related work is insufficient to establish a strong foundation.

Structure and Conformance: The manuscript generally conforms to PeerJ standards, and the sections flow logically. However, several parts that should be in the Introduction are misplaced in the Methodology section, and vice versa. Such structural issues disrupt the logical flow of the manuscript. Please refer to the attached PDF for detailed comments on the specific sections that need adjustment.

Introduction Clarity: The motivation for the study is adequately introduced, but it lacks strong support. Using only a single GAN model from 2020 to justify the limitations of current research does not provide enough motivation to support the proposed contribution. Adding more recent and diverse studies to highlight existing gaps would strengthen the motivation.

Formal Results and Definitions: The majority of the equations in the manuscript lack proper definitions for the symbols used, which, although sometimes understandable, could confuse readers. In particular, Equations 4, 5, and 6 seem to have inconsistencies in their definitions, which need clarification and standardization throughout the manuscript.

Experimental design

Scope and Technical Standard: The research content aligns with the journal's aims and scope, and the investigation appears to be conducted to a reasonable technical standard. However, the model only uses the ResNet family as the baseline, without any comparison to state-of-the-art (SOTA) models for Human Pose Estimation tasks. This is especially important given that ViTPose [1] achieves a Test AP of 93.3 on OCHuman. The current approach should demonstrate its benefits over existing SOTA models to establish its contribution. Furthermore, the authors claim that their dataset generation method is applicable across various datasets. However, during the creation of new masks, a fixed threshold is applied, which seems specifically designed to fit the dimensions of COCOA [2] images. This raises concerns regarding the applicability of this process to other datasets where image sizes may vary. To address this concern, it is recommended to include results from at least one other dataset to validate the generalizability of the approach.

Methods Description: The methods are described in considerable detail. However, the clarity and effectiveness of the proposed contribution could be improved with a more concise focus on key experimental elements and removing redundancy where appropriate.

Data Preprocessing: This section is described comprehensively, with all necessary details provided. However, aspects unrelated to the core contribution of the paper would be better placed in the Implementation Details section rather than in the Methodology section (refer to the attached PDF for specific comments).

Evaluation Methods and Metrics: The evaluation metrics, including AP (Average Precision) and related metrics, are adequately described. However, the inclusion of the Mean Squared Error (MSE) formula (Equation 7) is redundant, as MSE is a well-known metric in machine learning. It would be sufficient to mention MSE in text form, which would help to streamline the presentation and make the content less repetitive.

Citations and Justification: The citations provided are generally appropriate, but there are some critical gaps. For instance, in some parts of the manuscript, conclusions are presented without proper references, such as the statement "applying a mask could limit the ability to extract features for large-sized objects". Providing a reference or empirical data to support such claims would enhance the credibility and strength of the manuscript.

Experimental Details Placement: Some implementation-specific details, such as "terminating training after 54,000 iterations", are currently included in the Methodology section. These should be moved to a subsection titled "Implementation Details" to maintain a clear distinction between high-level experimental design and specific implementation settings, ensuring the Methodology remains focused on overarching concepts and design principles.

[1] Xu, Yufei, et al. "Vitpose: Simple vision transformer baselines for human pose estimation." Advances in Neural Information Processing Systems 35 (2022): 38571-38584.
[2] Zhu, Yan, et al. "Semantic amodal segmentation." Proceedings of the IEEE conference on computer vision and pattern recognition. 2017.

Validity of the findings

Impact and Novelty: The manuscript does not clearly assess the impact and novelty of the proposed approach. It is important to elaborate on how this work advances the current state-of-the-art, particularly in terms of robustness to occlusions compared to existing methods. The authors should provide a clear comparison of their method with relevant state-of-the-art models and highlight the specific contributions and improvements offered by their approach.

A significant concern is that the methodology focuses on reducing noise in publicly available datasets to improve model performance, which might limit the ability to test model generalization and robustness. While the automated pipeline for generating training data is a valuable contribution, the current mask generation process based on simple shapes (as shown in Figure 2) may not be sophisticated enough to convey the intended level of innovation or superiority.

Conclusions and Supporting Results: The conclusions are generally well stated and aligned with the supporting results. However, the discussion of the improvements achieved by the method could be more concrete, especially when presenting quantitative gains. The manuscript draws conclusions regarding the effectiveness of the method based solely on the significant improvement in AP-M, but does not delve into the reasons for the decline in performance in AP50, AP75, and AP-L metrics (see Table 1 and 2). AP-M calculates Average Precision for medium-sized objects, but no evidence is provided to support the claim that such objects are the key focus of Human Pose Estimation (HPE) tasks. Thus, the current conclusions on effectiveness require further justification.

Experiments and Evaluations: The experiments and evaluations performed are mostly satisfactory, but there is a significant gap in terms of using appropriate baselines. Specifically, the use of only ResNet models as baselines, without comparisons to state-of-the-art human pose estimation models (e.g., ViTPose), makes it challenging to evaluate the true competitiveness of the proposed method. Including comparisons with more advanced models is strongly recommended to validate the significance of the proposed approach. Additionally, the ablation study should provide deeper insights, yet the current discussion on epsilon values seems to offer little meaningful contribution to understanding model behavior. Furthermore, the cal_heat_map function in the provided code uses image and mask ratios that do not match those in the ablation study, which raises questions about the study's consistency. I expect the authors to provide a more insightful ablation analysis.

Identifying Unresolved Questions and Limitations: The conclusion does not adequately address unresolved questions or limitations. While some issues are mentioned in the Discussion, the analysis lacks sufficient evidence. Overemphasis on epsilon in the ablation study detracts from more significant model issues, such as the impact of poorly fitted mask contours on model performance. It is important to acknowledge challenges in adapting mask generation to datasets with varying image sizes and the potential negative impact of arbitrary mask shapes. Providing focused future research directions addressing these fundamental issues would add substantial value.

Additional comments

Please refer to the attached PDF for details.

Annotated reviews are not available for download in order to protect the identity of reviewers who chose to remain anonymous.
Cite this review as

---

## Round 0.2 · Major Revisions

· Academic Editor

Major Revisions

Dear authors,

After the previous revision round, some adjustments still need to be made. As a result, I once more suggest that you thoroughly follow the instructions provided by the reviewers to answer their inquiries clearly.

You are advised to critically respond to all comments point by point when preparing a new version of the manuscript and while preparing for the rebuttal letter. All the updates should be included in the new version of the manuscript.

Kind regards,
PCoelho

Reviewer 1 ·

Basic reporting

The revised manuscript is improved compared to the former version. My previous comments are well addressed, and the presentation is improved significantly. The composition pattern and some other ideas are well elaborated, making them clearer. Overall, I tend to accept this manuscript.

Experimental design

The revised manuscript is improved compared to the former version. My previous comments are well addressed, and the presentation is improved significantly. The composition pattern and some other ideas are well elaborated, making them clearer. Overall, I tend to accept this manuscript.

Validity of the findings

The revised manuscript is improved compared to the former version. My previous comments are well addressed, and the presentation is improved significantly. The composition pattern and some other ideas are well elaborated, making them clearer. Overall, I tend to accept this manuscript.

Additional comments

The revised manuscript is improved compared to the former version. My previous comments are well addressed, and the presentation is improved significantly. The composition pattern and some other ideas are well elaborated, making them clearer. Overall, I tend to accept this manuscript.

Cite this review as

Reviewer 2 ·

Basic reporting

The manuscript shows significant improvement in academic writing quality compared to the previous version; however, certain issues remain unaddressed. For detailed comments and necessary revisions, please refer to the notes provided in the attached PDF.

Experimental design

The comparison with the latest SOTA model for Human Pose Estimation, such as ViTPose [1], remains incomplete. I strongly recommend the authors demonstrate how their proposed method provides advantages over existing SOTA models to clearly establish its academic contribution.

In Table 2, the authors present a zero-shot testing approach. However, the reported improvements are minimal and, in many evaluation metrics, show no clear advantage over the baseline. I suggest the following:

Reassess the presentation of Table 2: Either revise the results to focus on training and testing on the OCHuman dataset to highlight the method's generalizability across datasets or remove the zero-shot results altogether.
Emphasize model generalization performance more clearly if this is a key claim, as it is currently underexplored in the manuscript.
This revision would ensure that the results align with the primary focus of the paper and provide a stronger justification for the method's effectiveness.

In addition, I would like to raise concerns regarding the Ablation Study:

Epsilon Values:

I do not fully understand the rationale for conducting experiments with epsilon values as small as
1e-7,1e-5. Given that the grayscale mask contains only three levels (0, 0.5, and 1), epsilon values of this magnitude seem smaller than the noise inherent in the dataset itself and are unlikely to produce meaningful insights.
Suggestion: Instead of focusing on epsilon values at such small scales, consider ablations that provide more practical comparisons.
More Meaningful Ablation Directions:
A more insightful ablation study could investigate the following:
The impact of using Figure 3(B)'s initial mask versus the refined mask on model training performance.
The effect of using alternative shapes (e.g., rectangles or other geometric forms) instead of ellipses for mask generation. (While the latter might slightly exceed the current scope of the paper, it could provide valuable insights into the robustness of the proposed approach.)
[1] Xu, Yufei, et al. "Vitpose: Simple vision transformer baselines for human pose estimation." Advances in Neural Information Processing Systems 35 (2022): 38571-38584.

Validity of the findings

The manuscript does not clearly assess the impact and novelty of the proposed approach. The authors should provide a clear comparison of their method with relevant state-of-the-art models and highlight the specific contributions and improvements offered by their approach.

A significant concern is that the methodology primarily focuses on reducing noise in publicly available datasets to improve model performance, which may limit the ability to evaluate model generalization and robustness. While the automated pipeline for generating training data is a valuable contribution, the current mask generation process, relying on simple geometric shapes, lacks sophistication and may not sufficiently demonstrate the intended level of innovation or superiority.

Additional comments

Please refer to the attached PDF for details.

Annotated reviews are not available for download in order to protect the identity of reviewers who chose to remain anonymous.
Cite this review as

---

## Round 0.3 · Minor Revisions

· Academic Editor

Minor Revisions

Dear authors,
Thanks a lot for your efforts to improve the manuscript.

Nevertheless, some concerns are still remaining that need to be addressed.
Like before, you are advised to critically respond to the remaining comments point by point when preparing a new version of the manuscript and while preparing for the rebuttal letter.

Kind regards,
PCoelho

Reviewer 2 ·

Basic reporting

The revised manuscript demonstrates notable improvements compared to the previous version. Many of the concerns raised in the earlier review have been addressed, and the overall presentation has been significantly enhanced. However, several refinements are still necessary to ensure clarity, consistency, and adherence to academic standards.

1. Typographical and Formatting Issues
Line 69: The full name Generative Adversarial Network does not match the abbreviation GAN, which should be corrected for consistency.
Line 226: "whichassist" is missing a space between words.
Line 227: "Here," is missing a space after the comma.
Similar minor typographical inconsistencies appear throughout the manuscript. A thorough proofreading is recommended to ensure consistency and clarity.
2. Mathematical Notation and Equation Formatting
Equations (5) and (6) could be rewritten using more standardized notation. Below is an improved LaTeX representation that aligns better with academic conventions:
"""
\begin{equation}
\mathcal{L}_\text{in} = \frac{\lambda_\text{in}}{|\mathcal{M}|} \sum_{i \in \mathcal{M}} \frac{1}{2} \left( \frac{\hat{y}_i - \mu}{\sigma} \right)^2,
\end{equation}
where \( \mathcal{M} \) denotes the set of indices corresponding to the masked region (\( m_i = 1 \)).

\begin{equation}
\mathcal{L}_\text{out} = \frac{\lambda_\text{out}}{|\mathcal{B}|} \sum_{i \in \mathcal{B}} \frac{1}{2} \left( \frac{\hat{y}_i - \mu}{\sigma} \right)^2,
\end{equation}
where \( \mathcal{B} \) denotes the background region (\( m_i = 0 \)).
"""
While this is a suggestion, adopting conventional notation would enhance clarity for readers.

Equations (3) and (7) require particular attention. The term "image" is not a scalar quantity. If the authors intend to represent element-wise multiplication between a scalar and a multi-channel or single-channel 2D matrix, the notation should be adjusted accordingly.

Explanation of symbols (Lines 208–212):
The description of mathematical symbols should maintain a consistent structure without additional delimiters that deviate from the formatting of other equations in the manuscript.

Notation Consistency:
If Height (H) and Width (W) are used in Equation (5), it is unclear why Equation (1) adopts a more cumbersome notation. Ensuring consistency across all equations would enhance clarity and facilitate comprehension.

Experimental design

While I acknowledge the authors’ justification for using 10e-7 and 10e-5 as the threshold for evaluating mask impact, the current ablation study lacks a clear turning point that demonstrates its effectiveness.
To rigorously validate that the selected ϵ is optimal, the authors should expand the ablation study to include at least ϵ=0.3,0.5,0.8. This would provide stronger empirical evidence to support the current choice and demonstrate whether it represents the best-performing setting.

Validity of the findings

The revised manuscript has addressed my previous comments for this part.

Additional comments

Several figures in the manuscript appear pixelated when zoomed in on the PDF.
If possible, I strongly recommend using Scalable Vector Graphics (SVG) instead of rasterized images (screenshots) to maintain clarity and readability.

Cite this review as

---

## Round 0.4 · accepted · Accept

· Academic Editor

Accept

Dear authors, we are pleased to verify that you meet the reviewer's valuable feedback to improve your research.

Thank you for considering PeerJ Computer Science and submitting your work.

Kind regards
PCoelho

Reviewer 2 ·

Basic reporting

The revised manuscript has addressed my previous comments for this part.

Experimental design

The revised manuscript has addressed my previous comments for this part.

Validity of the findings

The revised manuscript has addressed my previous comments for this part.

Additional comments

The term “1.6% improvement in AP” in the paper is misleading, as the improvement is in absolute percentage points, not relative percentage. Please revise this for clarity and correctness.

line 45 “the face(nose, eyes)...” is missing a space

Cite this review as